# Management compliance attention, social performance and pay gap: Evidence from China

**Fei Han[1], Yong Jiang[1]\*, Yanhan Sun[2]**

1 School of accounting, Shandong Management University, Jinan, Shandong, China, 2 School of Business and Administration, Shandong University of Finance and Economics; School of accounting, Shandong Management University. Erhuan east road, Lixia District, Jinan, Shandong, China

\* yongjiang_2024@163.com

## Abstract

The pronounced pay gap prevalent in companies has raised critical concerns regarding organizational equity. This study investigates mechanisms to mitigate pay gap through the lens of management compliance attention, with the dual objectives of enhancing intra-firm distributive justice and fostering sustainable organizational development. Furthermore, this study employs a comprehensive methodology, including textual analysis of annual reports to construct management compliance attention variables and the Chinese Huazheng ESG rating system to measure social performance. The results show that:(1) Management compliance attention is negatively correlated with pay gap($\beta$ = -0.010, p < 0.01); (2) Social performance is negatively correlated with pay gap($\beta$ = -0.035, p < 0.01); (3) Social performance strengthens the negative effect of management compliance attention on pay gap($\beta$ = -0.022, p < 0.05). Based on the results, we find that management compliance attention and social performance can narrow pay gap respectively, and that social performance positively moderates the relationship between compliance attention and pay gap. In addition, organizational resilience is identified as a partial mediator, explaining how compliance attention enhances firms' ability to adapt to external changes, thereby reducing pay gap. Heterogeneity analysis shows that management compliance attention and social performance effectively narrow pay gap in state-owned shareholding companies, and widen pay gap in private shareholding companies. This study contributes to the literature on the economic consequences of corporate compliance and social performance and the determinants of pay gap. Practical implications involve recommendations for firms to strengthen corporate compliance systems, enhance social performance and organizational resilience, and establish transparent compensation frameworks.

## 1. Introduction

Over the past decades, research on the executive-employee pay gap has proliferated, predominantly focusing on economic and behavioral patterns, institutional

**Data availability statement:** all relevant data are within the manuscript and its Supporting Information files.

**Funding:** This research was funded by the National Social Science Fund of China (Grant Number: 20BGL079) and the Shandong Provincial Social Science Planning Research Project (Grant Number: 24CFZJ16). The funders had no role in study design, data collection and analysis, decision to publish, or preparation of the manuscript.

**Competing interests:** The authors have declared that no competing interests exist.

determinants [1], human capital dynamics [2], rationality frameworks [3], and implications for corporate financial performance [4]. However, building on these theoretical foundations, a critical yet underexplored question emerges: whether management compliance attention can effectively mitigate pay gap.

Management attention reflects a strategic focus on specific organizational priorities, representing the rationale behind managerial decisions to undertake contextually appropriate actions at optimal times [5]. Drawing on the attention-based view (ABV), managerial cognition and action are fundamentally shaped by their embedded environments, which determine what aspects of the external landscape managers prioritize and which opportunities remain internalized within the firm [6,7]. Most growth theories posit that firm growth is contingent on the scope and limitations of management attention [8]. For instance, technology firms exhibited greater global expansion when management systematically integrated diverse external environmental factors into decision-making [9]. Similarly, top management attention to innovation has been shown to drive new product development [10], whereas attention to relationship marketing at trade shows enhances firm performance [11].

In China, the promulgation of compliance management policy documents has expanded the scope of corporate compliance obligations from state-owned enterprises (SOEs) to private firms. Amid the current complex and evolving economic landscape, corporate compliance serves as a critical mechanism for enhancing organizational risk prevention capabilities, improving competitiveness, and advancing social equity—fundamental prerequisites for lawful and sustainable business operations.Existing literature has predominantly examined the economic consequences of corporate compliance [12–14]. However, studies explicitly addressing management compliance attention remain scarce. This paper adopts a novel perspective by conceptualizing corporate compliance as an institutional outcome shaped by management compliance attention.

Existing research demonstrates that firms investing in pro-social initiatives exhibit stronger overall financial health [15,16]. Increasingly, companies are integrating social performance objectives into executive compensation frameworks [17]. Grounded in upper echelons theory, scholars have examined linkages between CEO characteristics (e.g., tenure, background) and social performance [18–21]. Concurrently, extensive literature focuses on the impact of corporate social responsibility (CSR) on firm performance [22–24]. However, scant attention has been paid to two critical gaps: (1) how social performance influences pay gaps, and (2) its role in mediating or moderating the relationship between management compliance attention and pay gap.

This study advances theoretical understanding of how compliance practices and social performance jointly shape pay equity while offering actionable insights for strengthening compliance culture, enhancing social performance, and designing transparent compensation systems.The main marginal contributions and innovations of this paper are as follows: Firstly, this study extends the application scenarios of behavioral theory and equity theory by introducing management compliance attention as a new antecedent variable in the analysis of pay gaps. By integrating behavioral theory's

emphasis on managerial decision-making heuristics with equity theory's focus on distributive justice perceptions, the paper constructs a novel theoretical framework that explains how management's strategic attention to compliance norms interacts with corporate social performance to influence internal pay equity. This contributes to the literature by demonstrating that compliance orientation not only serves as a regulatory response mechanism but also acts as a behavioral signal shaping organizational fairness perceptions. Secondly, The paper constructs management compliance attention variables through textual analysis, extracting keywords from the "Management's Discussion and Analysis" section of the annual reports of Chinese A-share listed companies. This provides a new quantitative method for measuring management compliance attention, which can more directly reflect the actual attention of management to compliance compared with previous studies. Thirdly, the study provides empirical evidence from China's institutional context, where regulatory pressures and stakeholder expectations are rapidly evolving. By examining state-owned and private enterprises separately, the paper reveals differential compliance-pay gap relationships driven by ownership characteristics, thus extending the generalizability of behavioral and equity theories to transitional economies. This contributes to the cross-cultural literature by identifying institutional moderators that shape compliance behavior's distributive consequences. The paper reveals that organizational resilience plays a partial mediating role in the relationship between management compliance attention and the pay gap. This enriches the application scope of organizational resilience theory and provides theoretical support for enterprises to optimize their pay structure by enhancing organizational resilience. The research also discovers that the impacts of management compliance attention and social performance on the pay gap vary under different ownership structures. In state-owned holding companies, they can effectively narrow the pay gap, while in private holding companies, they may widen it. This provides targeted references for different ownership enterprises to formulate reasonable pay policies and compliance strategies.

## 2. Literature review and hypothesis development

### 2.1. Management compliance attention and pay gap

While there is limited literature specifically examining the relationship between compliance attention and pay gap, existing research has explored the link between specific aspects of attention and pay disparities. For instance, from the perspective of pay equity, corporate management is urged to prioritize social equity [25]. The board's overall attention to information technology (IT) has been shown to reduce information asymmetry and significantly influence Chief Information Officer (CIO) compensation decisions [26]. The gender pay gap has also been a prominent area of research. Although college students are aware of the existence of gender pay gap, they frequently underestimate the magnitude of these disparities to a significant extent [27]. Analyst reports have been found to mitigate the gender pay gap by drawing stakeholders' attention to discriminatory practices [28]. Furthermore, increased attention to women's economic, social, and political rights has been linked to improvements in income, education, and support for future generations, thereby reducing overall pay gap [29]. Shifts in attention toward pay gap have been correlated with changes in post-tax and transfer inequality within countries [30]. Public concern over income inequality tends to rise during economic recessions, as observed in Europe and the United States [31]. However, individuals often prioritize access to healthcare and education over concerns about income inequality [32].

The concept of compliance originated in the early 20th century with the establishment of the U.S. Food and Drug Administration (FDA), marking the institutionalization of regulatory oversight. Its evolution accelerated significantly following the 2002 enactment of the Sarbanes-Oxley Act, which systematized compliance frameworks across industries [33]. Empirical evidence suggests that effective compliance enhances corporate competitive advantage and elevates reputational capital [34].To align with legal, regulatory, and ethical standards, firms must establish equitable compensation systems [35,36]. Standardized pay structures mitigate excessive pay gaps between executives and employees, thereby avoiding public scrutiny and internal discontent [37]. Furthermore, transparent disclosure of compensation criteria and processes—such as publishing pay distribution rationales—fosters employee understanding of equity mechanisms [38]. This practice not only reinforces compliance but also aligns with social performance objectives.

This study investigates the impact of management compliance attention on pay gap based on behavioral theory and equity theory.

**2.1.1. Behavioral theory perspective.** Behavioral theories posit that managers operate under cognitive limitations [39], managerial behavior within an organization is not only driven by economic incentives but also significantly influenced by social and psychological factors [40]. The frequent inclusion of keywords such as "compliance" and "supervision" in annual reports signals heightened management attention to regulatory adherence. First, the transparency of compliance mechanisms—including audit procedures and risk controls—mitigates employee concerns about opaque decision-making processes. Higher levels of management compliance attention correlate with greater emphasis on the transparency of compensation policies, thereby rationalizing pay gaps between executives and employees. Transparent compensation systems address institutional inequities, and compliance-oriented management is more likely to disclose executive-to-employee pay ratios, enhancing employee acceptance of reasonable disparity thresholds. Second, audit supervision and similar measures constrain managerial power abuse [41], reducing organizational inefficiencies and suppressing pay gaps. Compliance-focused management reinforces adherence to the "contribution-reward" matching principle, aligning perceptions of pay rationality between executives and non-managerial employees. When employees perceive compensation systems as meritocratic (i.e., "greater effort yields greater rewards"), they are more inclined to pursue higher returns through performance enhancement rather than counterproductive resistance, thereby curbing irrational pay disparities. Furthermore, excessive pay gaps heighten regulatory risks; firms with strong compliance awareness proactively control such disparities to avoid policy violations.

**2.1.2. Equity theory perspective: Compliance-driven external legitimacy pressures.** Equity theory primarily focuses on employees' perceptions of the ratio between their inputs and outcomes, as well as comparisons of this ratio with others [42]. The incentive effects of pay gaps depend on employees' perceived thresholds of fairness regarding distributive outcomes [43]. Organizational equity and collaboration mitigate psychological disparities among non-core executive team members and ordinary employees [44]. Management compliance attention optimizes pay gap fairness through governance structure enhancements. Internally, management's influence over boards significantly shapes pay disparities [45]. Under compliance-oriented management, the independence of compensation committees is strengthened, thereby curbing executives' self-determined remuneration. Externally, compliance-driven firms narrow pay gaps to avoid the negative labeling of "predatory pay" and prevent external investors from forming perceptions of organizational inequity.

Based on the preceding analysis and as illustrated in Fig 1, management compliance attention suppresses the pay gap through a dual-path mechanism:

H1.Management compliance attention is negatively correlated with pay gap.

## 2.2. Social performance and pay gap

Corporate social responsibility (CSR) enables firms to effectively manage and coordinate relationships with stakeholders [46]. For state-owned enterprises (SOEs) in China, fulfilling CSR and establishing equitable compensation contracts represent critical strategies for aligning organizational objectives with profit maximization. Empirical evidence suggests that SOEs' adherence to CSR significantly mitigates pay gaps between executives and ordinary employees [47]. Furthermore, CSR-driven accountability mechanisms incentivize transparency in compensation practices, reinforcing equity perceptions across hierarchical tiers.The social norms and personality traits are the two determinants of pay gap [48]. CEOs pay premiums are lower in companies with more social responsibility, and are more likely to be paid moderately to improve employee relationship and promote equity in income distribution [49]. Similarly, CEO incentives are negatively correlated with CSR performance [50]. CSR narrows the pay gap between executives and employees in companies following the innovation-oriented strategy [51]. Environmental tax reforms enhance environmental regulations, reduce the executive compensation and increase employee salary, hence narrow the pay gap [52].The appropriateness of compensation structure design affects the disclosure of ESG information through its impact on the design, operation and maintenance of

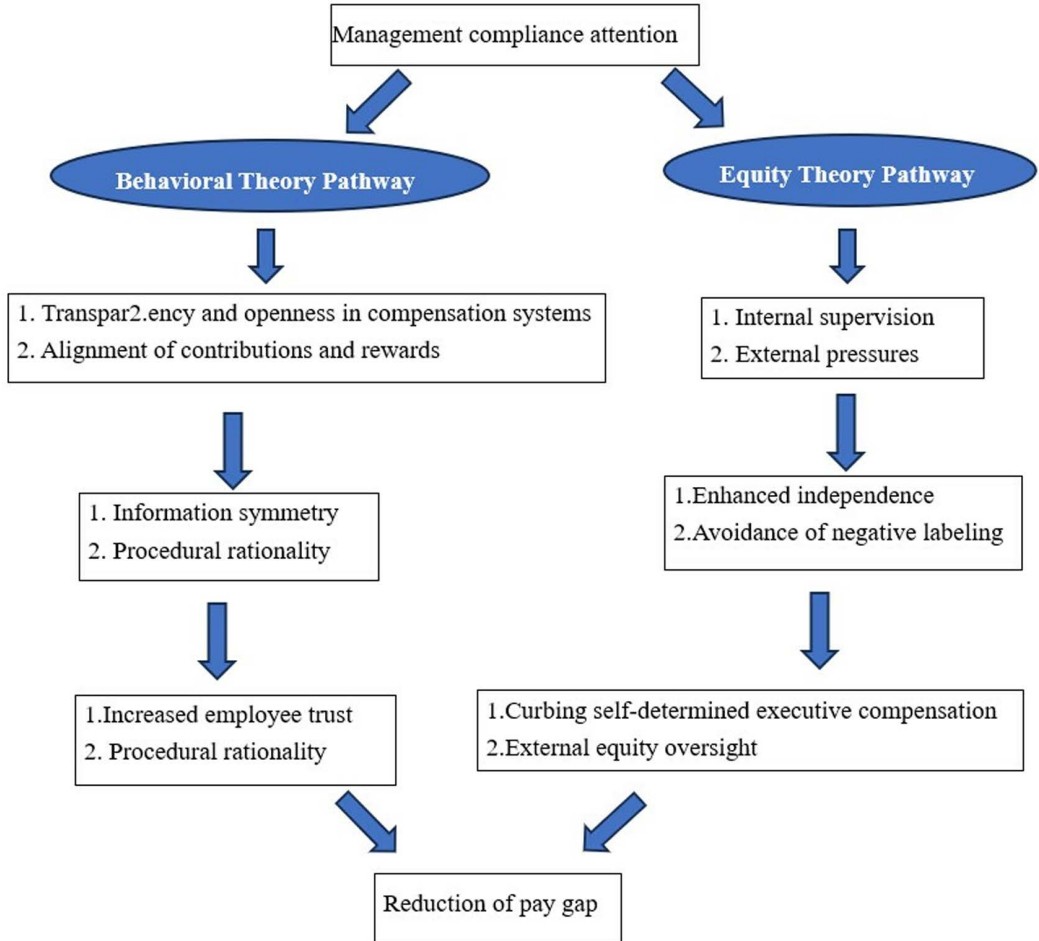

**Fig 1. Theoretical framework.** Therefore, the following hypothesis is proposed.

internal control [53]. Core executives have a key impact on ESG disclosure decisions in companies with large executive pay gap [54].The executive pay gap is positively correlated with the ESG disclosure [55].

Achieving social performance necessitates a rigorous examination of pay equity and the systematic advancement of fairness and social justice, both of which are integral to sustainable corporate development [56]. To attain high social performance, firms must prioritize rational compensation design and evidence-based structural validity. By optimizing compensation frameworks and implementing diversified incentive mechanisms, firms can narrow the pay gap between executives and employees.Relative deprivation theory posits that perceived inequities in distributive justice trigger feelings of injustice, whereas fairness in resource allocation fosters relational equity [42]. Pay equity not only boosts employee motivation and productivity but also strengthens organizational innovation capabilities and market competitiveness. Specifically, equitable compensation systems mitigate counterproductive comparisons among employees, aligning individual aspirations with collective goals.

Corporate social performance (CSP) reflects a firm's commitment to ethical standards and social responsibility [57]. The extent to which firms fulfill these responsibilities significantly shapes employee attitudes and behaviors [58]. To align with CSP principles, firms must proactively safeguard employee well-being, promote social equity, and integrate ethical imperatives into compensation systems. This entails avoiding exploitative practices—such as excessive executive

self-enrichment or infringement of employee rights—which undermine organizational legitimacy. Strengthening social responsibility and ethical governance not only narrows pay gaps but also enhances corporate reputation and stakeholder trust [59]. CSP further emphasizes symbiotic growth between firms and their employees. By investing in skill development programs, career advancement opportunities, and innovation-enabling resources, firms empower employees to realize their potential while fostering equitable value creation. This shared-growth philosophy inherently reduces pay gaps by aligning individual and organizational success.

A firm, as a nexus of stakeholders including shareholders, employees, consumers, and communities [60], must balance competing interests to achieve high CSP. Prioritizing social performance necessitates a heightened focus on employee welfare. To attract and retain talent while boosting satisfaction and loyalty, firms increasingly adopt equitable compensation frameworks [61]. Equity theory warns that excessive pay gaps risk triggering perceptions of distributive injustice, eroding employees' organizational identification and belongingness. Such perceptions can cascade into reduced productivity and reputational damage [42]. Consequently, CSP-driven firms strategically moderate pay gaps by embedding fairness principles into compensation design—for instance, uplifting baseline remuneration for non-executive staff—to ensure alignment with collective equity norms.Additionally, robust CSP enhances corporate reputation—a critical intangible asset [62]. Firms prioritizing short-term profits at the expense of employee rights risk reputational erosion. To mitigate this, CSP-oriented firms institutionalize pay gap controls within ethically bounded ranges, signaling commitment to employee welfare through transparent compensation policies.

Therefore, the following hypothesis is proposed:

H2. Social performance is negatively correlated with pay gap.

## 2.3. Management compliance attention, social performance and pay gap

Recent studies have sought to explain why and how social performance is linked to financial performance by identifying moderating and mediating variables [23,63,64]. Social performance plays a moderating role in the process through which financial risk negatively impacts financial performance [65]. Corporate social performance requires firms to adhere to ethical standards in their operations [66]. Within the framework of social performance, management compliance attention can incentivize firms to optimize incentive mechanisms, ensuring that employee compensation aligns closely with individual contributions. This alignment fosters employee motivation and creativity while mitigating the expansion of pay disparities [67]. Following the implementation of CSR regulations, firms adopting innovation-oriented strategies have responded more proactively, narrowing pay gaps by improving employee relations and achieving disproportionately higher growth in average employee compensation relative to executive pay [51].

While managerial self-interest may incentivize inflated pay gaps to maximize personal gains or assert hierarchical dominance, management compliance attention counteracts such agency problems. Compliance-oriented governance restrains self-serving behaviors, promoting equitable compensation policies that reduce pay disparities. The amplifying role of social performance in this relationship operates through two distinct yet interconnected mechanisms. First, firms demonstrating robust social performance exhibit heightened managerial prudence in addressing pay inequities, actively curtailing pay gaps within compliance frameworks to balance stakeholder interests and safeguard corporate reputation. Second, signaling theory elucidates how management compliance attention communicates adherence to governance norms and fairness principles, fostering employee trust and market confidence [68]. This signaling effect aligns with equity theory, which underscores that equitable compensation practices mitigate perceptions of distributive injustice, thereby strengthening organizational identification and cohesion [42]. Together, these mechanisms highlight how social performance reinforces compliance-driven governance, ensuring pay equity while enhancing both internal legitimacy and external competitiveness.

For firms with strong social performance, fairness expectations among employees are elevated. Under compliance attention, management optimizes compensation systems to align with social responsibility principles, ensuring pay gaps

remain within ethically defensible bounds. This dual focus on compliance and equity not only strengthens employee perceptions of fairness but also bolsters corporate competitiveness and reputation.

Therefore, this study proposes the following hypothesis

H3.The social performance strengthens the negative impact of management compliance attention on pay gap.

## 3. Methodology

### 3.1. Data collection

The initial dataset is sourced from the China Stock Market Accounting Research (CSMAR) database. This study employs a sample of Chinese A-share listed companies spanning 2010–2022, excluding financial firms, ST/*ST companies, and observations with missing values. To mitigate the influence of extreme values, all continuous variables are winsorized at the 1% and 99% percentiles. Following these procedures, the final sample comprises 24,708 firm-year observations.

### 3.2. Variables

**3.2.1. Dependent variable.** The dependent variable is the pay gap, measured as the compensation disparity between DSEs (directors, supervisors, and executives) and ordinary employees. Based on the relevant research methods [69], we calculate pay gap using the following formula:

$$Pay\_gap = \frac{average\ \text{DSE}\ pay}{average\ employee\ pay} \tag{1}$$

$$average\ \text{DSE}\ pay = \frac{total\ \text{DSE}\ pay}{number\ of\ \text{DSE}} \tag{2}$$

$$average\ employee\ pay = \frac{total\ pay - \text{DSE}\ pay}{number\ of\ employees - number\ of\ \text{DSE}} \tag{3}$$

**3.2.2. Independent variable.** The independent variable, management compliance attention, is constructed based on keyword frequency in the "Management's Discussion and Analysis" (MD&A) sections of annual reports. Drawing on prior studies of corporate compliance [70–72], we identify 11 keywords: compliance, lawfulness, regulation, law, rule, norm, auditing, supervision, risk management, information security, and data protection. The variable is operationalized as:

$$Management\ Compliance\ Attention = \ln(Keyword\ Count + 1) \tag{4}$$

where Keyword Count represents the annual frequency of these terms in MD&A disclosures.

**3.2.3. Moderating variable.** Social performance, is measured using the Huazheng ESG rating system, a comprehensive framework evaluating the fulfillment of social responsibility by Chinese A-share listed companies across five dimensions: human capital development, product responsibility, supply chain sustainability, social contribution, and data security and privacy governance. Higher ratings indicate stronger alignment with social responsibility objectives, with the index synthesizing these dimensions to quantify firms' commitment to ethical and sustainable practices.

**3.2.4. Control variables.** We include firm-level controls to account for confounding factors. Detailed definitions and operationalizations of all variables are provided in Table 1.

### 3.3. Empirical model

Drawing on the methodological framework [51], the following econometric model is specified to empirically examine the hypothesized relationships:

$$Pay\_gap_{it} = \beta_0 + \beta_1 Com\_att_{it} + \beta_2 Social_{it} + \beta_3 Com\_att_{it} \times Social_{it} + \beta_4 Employee_{it}$$
$$+ \beta_5 Lev_{it} + \beta_6 Revgrowth_{it} + \beta_7 Div\_ratio_{it} + \beta_8 Taturn_{it} + \beta_9 Indep\_ratio_{it}$$
$$+ \beta_{10} Top3\_pay_{it} + \sum Industry + \sum Year + \varepsilon_{it}$$

(5)

## 4. Results

### 4.1. Descriptive statistics

In Table 2, the mean value of Pay_gap is 0.987. When converted back to its pre-natural logarithm form, the mean is 2.68, indicating that the average compensation of DSEs (directors, supervisors, and executives) is 2.68 times that of ordinary employees. This suggests the presence of significant pay gaps within the sample firms. The standard deviation of Pay_gap is 0.653, reflecting variations in pay gaps across different sample firms.The mean value of Com_att is 1.913. When converted back to its pre-natural logarithm form, the mean is 6.77, indicating that compliance-related terms appear 6.77 times on average in the annual reports of the sample firms. This demonstrates a certain level of management compliance attention among the sample firms. The standard deviation of Com_att is 0.923, suggesting variability in the degree of compliance attention across different sample firms. Social performance (Social) exhibits a mean of 1.402 (SD = 0.311), indicating variability in firms' commitment to social responsibility. The natural logarithm of employee count (Employee) averages 7.721, corresponding to approximately

**Table 1. Variable description.**

| Variable type | Variable name | Definition |
|---|---|---|
| Dependent variable | Pay_gap | Natural logarithm of (Average directors,supervisors and executives pay divided by the average employee pay) |
| Independent variable | Com_att | Natural logarithm of (number of management compliance attention key words frequency + 1) |
| Moderator variable | Social | Social rating rank of Chinese Huazheng ESG rating system |
| | Employee | Natural logarithm of employees number |
| | Lev | Total debt divided by total assets |
| | Revgrowth | (Current year's revenue- Prior year's revenue)/ Current year's revenue |
| Control variables | Div_ratio | Payout ratio |
| | Taturn | Revenue divided by total assets |
| | Indep_ratio | Number of independent directors divided by Number of Board members |
| | Top3_pay | Natural logarithm of Top 3 executives pay |
| | Year | 2010–2022 |
| | Industry | China Securities Regulatory Commission 2012 indsutry classification |

**Table 2. Descriptive Statistics.**

| variable | obs | mean | stddev | min | max |
|---|---|---|---|---|---|
| Pay_gap | 24708 | 0.987 | 0.653 | -0.714 | 2.754 |
| Com_att | 24708 | 1.913 | 0.923 | 0.000 | 4.220 |
| Social | 24708 | 1.402 | 0.311 | 0.000 | 1.792 |
| Employee | 24708 | 7.721 | 1.212 | 5.075 | 11.128 |
| Lev | 24708 | 0.405 | 0.196 | 0.053 | 0.851 |
| Revgrowth | 24708 | 0.201 | 0.390 | -0.426 | 2.537 |
| Div_ratio | 24708 | 0.273 | 0.306 | 0 | 1.829 |
| Taturn | 24708 | 0.628 | 0.405 | 0.1 | 2.454 |
| Indep_ratio | 24708 | 0.376 | 0.053 | 0.333 | 0.571 |
| Top3_pay | 24708 | 14.514 | 0.713 | 12.794 | 16.524 |

2,255 employees per firm. Leverage (Lev) demonstrates a mean ratio of 0.405 (SD = 0.201), reflecting an average debt-to-asset ratio of 40.5%. Revenue growth (Revgrowth) averages 0.201, suggesting a 20.1% year-on-year expansion. Dividend payout ratio (Div_ratio) stands at 0.273, implying that 27.3% of net profits are distributed to shareholders. Independent director representation (Indep_ratio) averages 0.376, consistent with the 37.6% board independence mandated by Chinese corporate governance regulations. Finally, the natural logarithm of top-three executive compensation (Top3_pay) averages 14.514, equivalent to approximately ¥2,071,948 annually, with substantial cross-firm variation (SD = 0.876).

## 4.2. Correlated coefficient matrix analysis

As shown in Table 3, the correlation coefficient between management compliance attention (Com_att) and pay gap (Pay_gap) is −0.023 (significant at the 1% level), providing preliminary empirical support for H1. This negative association aligns with the hypothesis that heightened compliance focus curbs excessive executive-employee compensation disparities.However, the correlation coefficient between social performance (Social) and Pay_gap is 0.118, which contradicts H2. This counterintuitive result may stem from omitted variable bias (e.g., unobserved firm-specific governance practices) or measurement limitations in the social performance construct, necessitating rigorous multivariate regression analysis to disentangle confounding effects.Notably, the maximum pairwise correlation among all variables is 0.426 (significant at the 1% level), well below the conventional multicollinearity threshold of 0.7, ensuring the robustness of subsequent regression estimates.

## 4.3 Regression analysis

Table 4 presents three models using data from 2010 to 2022. In M4, the coefficient of Com_att is −0.010 (significant at the 1% level), indicating that higher level of management compliance attention is associated with smaller pay gap, thereby supporting H1. The coefficient of Social is −0.035 (significant at the 1% level), suggesting that improved social performance reduces pay gaps, thus validating H2.The interaction term coefficient (Com_att × Social) is −0.022 (significant at the 5% level), implying that social performance amplifies the negative effect of management compliance attention on pay gaps, which provides robust empirical evidence for H3.

## 5. Robust test

### 5.1 Marginal effects analysis

To further investigate the interplay among management compliance attention, social performance, and pay gap, the sample is bifurcated into high- and low-social performance groups using a cutoff of one standard deviation above and below the

**Table 3. Correlation Matrix.**

| | 1 | 2 | 3 | 4 | 5 | 6 | 7 | 8 | 9 | 10 |
|---|---|---|---|---|---|---|---|---|---|---|
| 1.Pay_gap | 1.000 | | | | | | | | | |
| 2.Com_att | −0.023*** | 1.000 | | | | | | | | |
| 3.Social | 0.118*** | 0.020*** | 1.000 | | | | | | | |
| 4.Employee | 0.426*** | −0.016** | 0.141*** | 1.000 | | | | | | |
| 5.Lev | 0.040*** | −0.033*** | −0.034*** | 0.400*** | 1.000 | | | | | |
| 6.Revgrowth | 0.006 | 0.025*** | −0.035*** | −0.017*** | 0.067*** | 1.000 | | | | |
| 7.Div_ratio | 0.073*** | −0.011* | 0.065*** | 0.004 | −0.173*** | −0.100*** | 1.000 | | | |
| 8.Taturn | 0.126*** | 0.037*** | 0.001 | 0.259*** | 0.184*** | 0.057*** | −0.011* | 1.000 | | |
| 9.Indep_ratio | 0.001 | −0.028*** | 0.064*** | −0.015** | −0.011* | 0.000 | −0.006 | −0.020*** | 1.000 | |
| 10.Top3_pay | 0.611*** | −0.120*** | 0.159*** | 0.356*** | 0.138*** | 0.026*** | −0.001 | 0.104*** | 0.013** | 1.000 |

Note: *** $p < 0.01$, ** $p < 0.05$, * $p < 0.1$

**Table 4. Regression results.**

| VARIABLES | M1 Pay_gap | M2 Pay_gap | M3 Pay_gap | M4 Pay_gap |
|---|---|---|---|---|
| Com_att | −0.011*** | | −0.010*** | −0.010*** |
| | (−2.888) | | (−2.805) | (−2.811) |
| Social | | −0.034*** | −0.033*** | −0.035*** |
| | | (−3.547) | (−3.479) | (−3.632) |
| Com_att × Social | | | | −0.022** |
| | | | | (−2.165) |
| Employee | 0.156*** | 0.158*** | 0.158*** | 0.158*** |
| | (52.265) | (52.006) | (52.063) | (52.041) |
| Lev | −0.367*** | −0.375*** | −0.376*** | −0.376*** |
| | (−20.146) | (−20.361) | (−20.438) | (−20.459) |
| Revgrowth | 0.016** | 0.016** | 0.016** | 0.016** |
| | (2.252) | (2.153) | (2.189) | (2.161) |
| Div_ratio | 0.035*** | 0.038*** | 0.038*** | 0.037*** |
| | (3.696) | (3.951) | (3.913) | (3.892) |
| Taturn | −0.079*** | −0.080*** | −0.080*** | −0.080*** |
| | (−9.480) | (−9.520) | (−9.523) | (−9.520) |
| Indep_ratio | 0.082 | 0.095* | 0.095* | 0.094* |
| | (1.556) | (1.796) | (1.789) | (1.773) |
| Top3_pay | 0.574*** | 0.576*** | 0.576*** | 0.576*** |
| | (118.536) | (118.537) | (118.524) | (118.523) |
| Year | controlled | controlled | controlled | controlled |
| Industry | controlled | controlled | controlled | controlled |
| Constant | −8.028*** | −8.059*** | −8.022*** | −8.087*** |
| | (−101.410) | (−103.233) | (−101.321) | (−102.240) |
| Observations | 24,708 | 24,708 | 24,708 | 24,708 |
| R-squared | 0.561 | 0.561 | 0.561 | 0.561 |

Note:t-statistics in parentheses, *** p<0.01, ** p<0.05, * p<0.1

mean. As depicted in Fig 2, the regression slope between management compliance attention and pay gap is markedly steeper in the high-social performance cohort. This result demonstrates that elevated social performance significantly amplifies the suppressive effect of management compliance attention on pay gaps, thereby providing robust empirical support for H3.

## 5.2 Replacement of variables

In M1 of Table 5, we excluded the keyword "law" from the 11 keywords constituting Com_att, added 1 to the keyword count, and took the natural logarithm to create the updated management compliance attention variable, Com_attrb. The coefficient of Com_attrb is -0.008, significant at the 5% level, indicating that even without considering management's attention to legal matters, management compliance attention still exerts a significant negative impact on pay gaps, thereby supporting H1. The coefficient of Social is -0.035, significant at the 1% level, validating H2. The interaction term coefficient is -0.021, significant at the 5% level, providing evidence for H3.In M2, the variable Social is replaced by S_score, derived from the ESG rating system of CNRDS (Chinese Research Data Services Platform). The coefficient of the updated social performance variable, S_score, is -0.017, significant at the 1% level, supporting H2. The interaction term coefficient between S_score and the management compliance attention variable, Com_attd, is -0.017, significant at the 1% level, which validates H3.

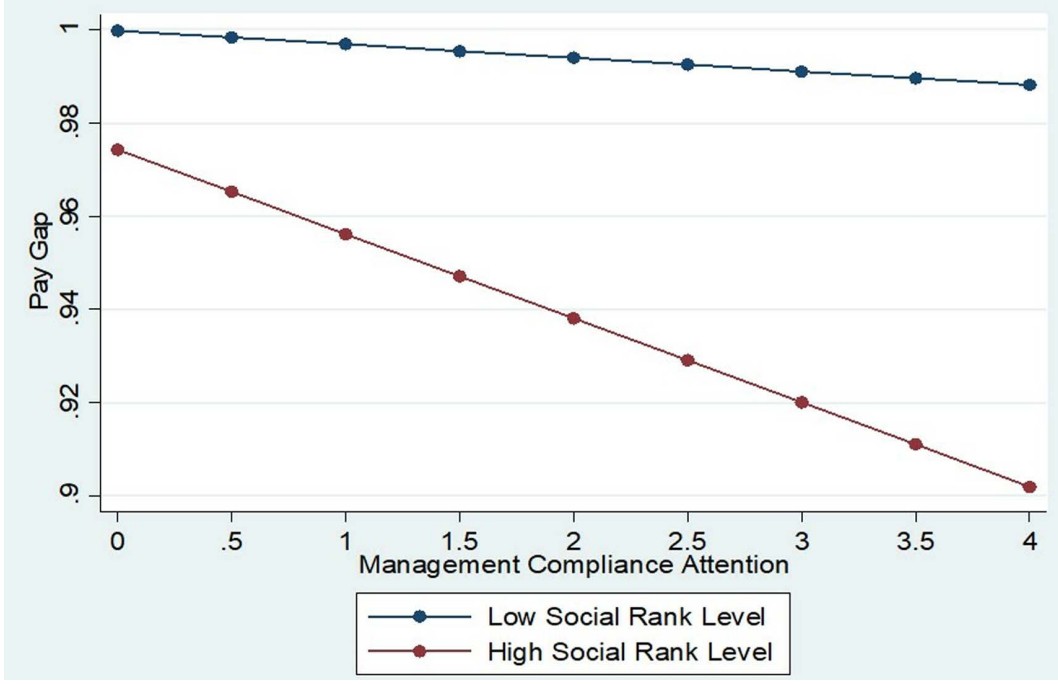

**Fig 2. Marginal effects analysis.**

### 5.3 Endogeneity issue

The baseline regression results indicate that management compliance attention effectively reduces the pay gap. However, the pay gap may reciprocally influence management compliance attention, suggesting potential bidirectional causality and endogeneity concerns. To address this issue, Column M1 of Table 6 implements a one-period lag for all variables except year and industry fixed effects. The coefficient estimate for l.Com_att (–0.009) is statistically significant at the 5% level, confirming H1. The coefficient for l.Social (–0.048) is significant at the 1% level, verifying H2, while the interaction term coefficient (–0.022) reaches significance at the 10% level, supporting H3. Columns M2 and M3 of Table 6 adopt an instrumental variable (IV) approach, using one-year lagged management compliance attention as the instrument and applying two-stage least squares (2SLS). The first-stage results reveal a coefficient of 0.611 for l.Com_att (p < 0.01) and an adjusted $R^2$ of 0.493, indicating strong instrument relevance and model fit. In the second stage, the coefficient for Com_attpr1 (-0.012) is significant at the 10% level, Social (-0.050) at the 1% level, and the interaction term (-0.058) at the 1% level. These results confirm that the original H1-H3 remain valid after addressing endogeneity.

### 5.4 Further analysis

Ownership structure heterogeneity may lead to differential compliance attention and pay gap management practices. Column M1 of Table 7 presents results for state-owned holding companies. The coefficient estimate for Com_att (–0.019) is significant at the 1% level (p < 0.01), while Social (–0.121) and the interaction term (-0.034) achieve significance at the 1% and 5% levels respectively. These findings demonstrate that in state-owned listed firms, both management compliance attention and social performance independently reduce pay gaps, with social performance amplifying the negative effect of compliance attention (supporting H3).

   In contrast, Column M2 results for private holding companies reveal a positive Com_att coefficient (0.009, p < 0.05) and Social coefficient (0.037, p < 0.01), while the interaction term (-0.007) remains insignificant. This indicates that

**Table 5. Replacement of variables.**

| VARIABLES | M1 Pay_gap | M2 Pay_gap |
|---|---|---|
| Com_attrb | −0.008** | |
| | (−2.176) | |
| Social | −0.035*** | |
| | (−3.647) | |
| Com_attrb×Social | −0.021** | |
| | (−2.109) | |
| Com_att | | −0.010*** |
| | | (−2.588) |
| S_score | | −0.017*** |
| | | (−3.060) |
| Com_att×S_score | | −0.017*** |
| | | (−3.364) |
| Employee | 0.158*** | 0.157*** |
| | (52.011) | (52.407) |
| Lev | −0.376*** | −0.366*** |
| | (−20.445) | (−20.085) |
| Revgrowth | 0.016** | 0.016** |
| | (2.155) | (2.215) |
| Div_ratio | 0.037*** | 0.036*** |
| | (3.898) | (3.763) |
| Taturn | −0.080*** | −0.080*** |
| | (−9.526) | (−9.580) |
| Indep_ratio | 0.094* | 0.078 |
| | (1.781) | (1.483) |
| Top3_pay | 0.576*** | 0.575*** |
| | (118.518) | (118.615) |
| Year | controlled | controlled |
| Industry | controlled | controlled |
| Constant | −8.090*** | −8.068*** |
| | (−102.202) | (−102.942) |
| Observations | 24,708 | 24,708 |
| R-squared | 0.561 | 0.561 |

management compliance attention and social performance in private firms fail to narrow pay gaps and may even exacerbate them. A plausible explanation lies in the superficial compliance practices often observed in private enterprises, where compliance attention may exist primarily in policy documents or rhetorical commitments rather than meaningful integration into compensation systems or operational processes. Consequently, even when present, compliance measures may lack the institutional rigor necessary to counteract pay inequality.

## 5.5 The mediating role of organizational resilience

Drawing on dynamic capabilities theory, firms must integrate and allocate internal/external resources while developing adaptive capacities to navigate changing market environments and foster sustainable growth [73,74]. Organizational resilience, a critical dynamic capability, emphasizes rapid processing of and responsiveness to external shocks. First, prioritizing

**Table 6. Endogenous issue.**

| VARIABLES | M1 | M2 | M3 |
|---|---|---|---|
|  |  | first stage | second stage |
|  | Pay_gap | Com_att | Pay_gap |
| I.Com_att | −0.009** |  |  |
|  | (−1.992) |  |  |
| I.Social | −0.048*** |  |  |
|  | (−3.871) |  |  |
| I.Com_att × I.Social | −0.022* |  |  |
|  | (−1.722) |  |  |
| Com_attpr1 |  |  | −0.012* |
|  |  |  | (−1.762) |
| Social |  |  | −0.050*** |
|  |  |  | (−4.321) |
| Com_attpr1 × Social |  |  | −0.058*** |
|  |  |  | (−3.125) |
| I.Com_att |  | 0.611*** |  |
|  |  | (103.791) |  |
| I.Employee | 0.157*** | 0.013*** |  |
|  | (40.916) | (2.624) |  |
| I.Lev | −0.343*** | −0.120*** |  |
|  | (−14.824) | (−4.083) |  |
| I.Div_ratio | 0.024** | −0.008 |  |
|  | (1.973) | (−0.487) |  |
| I.Revgrowth | 0.000** | 0.000 |  |
|  | (2.406) | (0.610) |  |
| I.Taturn | −0.054*** | −0.001 |  |
|  | (−5.154) | (−0.102) |  |
| I.Indep_ratio | 0.055 | −0.021 |  |
|  | (0.821) | (−0.248) |  |
| I.Top3_pay | 0.510*** | −0.004 |  |
|  | (83.941) | (−0.532) |  |
| Employee |  |  | 0.156*** |
|  |  |  | (44.099) |
| Lev |  |  | −0.338*** |
|  |  |  | (−15.607) |
| Revgrowth |  |  | 0.031*** |
|  |  |  | (3.388) |
| Div_ratio |  |  | 0.036*** |
|  |  |  | (3.309) |
| Taturn |  |  | −0.094*** |
|  |  |  | (−9.580) |
| Indep_ratio |  |  | 0.077 |
|  |  |  | (1.270) |
| Top3_pay |  |  | 0.582*** |
|  |  |  | (104.134) |
| Year | controlled | controlled | controlled |

*(Continued)*

**Table 6.** (Continued)

| VARIABLES | M1 | M2 | M3 |
|---|---|---|---|
| | | first stage | second stage |
| | Pay_gap | Com_att | Pay_gap |
| Industry | controlled | controlled | controlled |
| Constant | −7.137*** | 0.007 | −8.163*** |
| | (−70.668) | (0.058) | (−88.475) |
| Observations | 18,341 | 18,341 | 18,341 |
| R–squared | 0.492 | 0.493 | 0.575 |
| F | 186.2 | 190.9 | 260.2 |

compliance as a foundational strategy reduces non-compliance risks and strengthens risk resilience. By establishing robust compliance systems and conducting proactive risk assessments, firms enhance operational robustness and risk resistance. This process improves resource allocation flexibility and strengthens adaptive capacities for future challenges [75,76]. Second, management compliance attention enables firms to align pay gap management with strategic objectives while effectively responding to market changes. This alignment enhances employee acceptance of compensation structures. Moreover, compliance-driven organizational resilience reinforces the effectiveness of flexible, sustainable pay strategies.

Based on the relevant research methods [77], we operationalize the following model to test the mediating role of organizational resilience:

$$
\begin{aligned}
Pay\_gap_{it} = {}& \beta_0 + \beta_1 \times Com\_att_{it} + \beta_2 Employee_{it} + \beta_3 Lev_{it} \\
& + \beta_4 Revgrowth_{it} + \beta_5 Div\_ratio_{it} + \beta_6 Taturn_{it} \\
& + \beta_7 Indep\_ratio_{it} + \beta_8 Top3\_pay_{it} + \sum Industry + \sum Year + \varepsilon_{it}
\end{aligned}
\tag{6}
$$

$$
\begin{aligned}
Orgre_{it} = {}& \beta_0 + \beta_1 \times Com\_att_{it} + \beta_2 Employee_{it} + \beta_3 Lev_{it} \\
& + \beta_4 Revgrowth_{it} + \beta_5 Div\_ratio_{it} + \beta_6 Taturn_{it} \\
& + \beta_7 Indep\_ratio_{it} + \beta_8 Top3\_pay_{it} \\
& + \sum Industry + \sum Year + \varepsilon_{it}
\end{aligned}
\tag{7}
$$

$$
\begin{aligned}
Pay\_gap_{it} = {}& \beta_0 + \beta_1 \times Com\_att_{it} + \beta_2 \times Orgre_{it} + \beta_3 Employee_{it} \\
& + \beta_4 Lev_{it} + \beta_5 Revgrowth_{it} + \beta_6 Div\_ratio_{it} + \beta_7 Taturn_{it} \\
& + \beta_8 Indep\_ratio_{it} + \beta_9 Top3\_pay_{it} + \sum Industry + \sum Year + \varepsilon_{it}
\end{aligned}
\tag{8}
$$

Organizational resilience (Orgre) is operationalized in Equation (6) using two-dimensional framework: long-term growth and financial volatility [78]. Long-term growth is measured by the three-year cumulative operating income growth rate of sample firms, while financial volatility is calculated as the standard deviation of monthly stock returns over a 12-month period. After data standardization, the entropy weighting method is employed to compute a composite organizational resilience score.

Table 8 presents the mediation analysis results. Column M1 shows a statistically significant positive association between management compliance attention (Com_att) and organizational resilience ($\beta = 0.00068$, $p < 0.01$). Column M2 reveals a significant negative relationship between Com_att and pay gap ($\beta = -0.01795$, $p < 0.01$). In Column M3, the Com_att coefficient remains negative and significant ($\beta = -0.01705$, $p < 0.01$), with a slight increase in magnitude compared to Column M2. Notably, organizational resilience demonstrates a significant negative effect on pay gap ($\beta = -0.028$, $p < 0.01$). The Sobel test yields a z-statistic of -3.02 ($p = 0.003$), confirming partial mediation of the Com_att-pay gap relationship through organizational resilience.

**Table 7. Group analysis.**

| VARIABLES | M1 | M2 |
|---|---|---|
| | state-owned holding | private holding |
| | Pay_gap | Pay_gap |
| Com_att | −0.019*** | 0.009** |
| | (−3.154) | (2.177) |
| Social | −0.121*** | 0.037*** |
| | (−7.378) | (3.476) |
| Com_att × Social | −0.034** | −0.007 |
| | (−2.017) | (−0.650) |
| Employee | 0.166*** | 0.195*** |
| | (34.894) | (52.357) |
| Lev | −0.246*** | −0.255*** |
| | (−8.139) | (−11.998) |
| Revgrowth | −0.051*** | 0.005 |
| | (−4.223) | (0.604) |
| Div_ratio | −0.030* | 0.040*** |
| | (−1.701) | (3.889) |
| Taturn | −0.037*** | −0.091*** |
| | (−2.899) | (−8.890) |
| Indep_ratio | 0.004 | 0.094 |
| | (0.050) | (1.536) |
| Top3_pay | 0.633*** | 0.534*** |
| | (79.007) | (95.723) |
| Constant | −8.932*** | −7.823*** |
| | (−72.057) | (−80.265) |
| Observations | 9,828 | 14,880 |
| R-squared | 0.604 | 0.607 |

## 6. Discussion

The findings of this study provide valuable insights into the interrelationships among management compliance attention, social performance, and the pay gap within Chinese A - share listed companies. The results reveal a negative correlation between management compliance attention and the pay gap. This implies that when companies prioritize compliance, the compensation disparity between directors, supervisors, and executives (DSE) and that of ordinary employees is likely to decrease. This result is in line with the Behavioral Theory and the Equity Theory, which propose that management's focus on compliance can foster more equitable and transparent compensation practices, thereby narrowing the pay gap.

Furthermore, the study demonstrates that social performance significantly moderates the relationship between management compliance attention and the pay gap. Specifically, higher levels of social performance strengthen the negative impact of management compliance attention on the pay gap. This suggests that companies with outstanding social performance are more inclined to incorporate compliance into their compensation systems and daily operations, leading to a more equitable distribution of pay. This finding is consistent with the Stakeholder Theory, which emphasizes the significance of fulfilling social responsibilities and upholding ethical standards in corporate operations. By doing so, companies can enhance their corporate reputation and employee loyalty, ultimately contributing to a more sustainable and just compensation structure.

The mechanism analysis further indicates that organizational resilience partially mediates the relationship between management compliance attention and the pay gap. This suggests that companies with greater management compliance

**Table 8. Mediation effect.**

| VARIABLES | M1 | M2 | M3 |
|---|---|---|---|
| | Orgre | Pay_gap | Pay_gap |
| Orgre | | | −1.32578*** |
| | | | (−11.77925) |
| Com_att | 0.00068*** | −0.01795*** | −0.01705*** |
| | (3.12491) | (−4.66606) | (−4.44390) |
| Employee | 0.00357*** | 0.15305*** | 0.15778*** |
| | (21.41425) | (51.76489) | (53.02433) |
| Lev | 0.00191** | −0.53071*** | −0.52818*** |
| | (1.98311) | (−31.01421) | (−30.94948) |
| Revgrowth | −0.00782*** | 0.01937** | 0.00900 |
| | (−17.75002) | (2.47822) | (1.14709) |
| Div_ratio | 0.00344*** | 0.06619*** | 0.07076*** |
| | (5.95514) | (6.45352) | (6.91300) |
| Taturn | −0.00106** | 0.01908** | 0.01768** |
| | (−2.42826) | (2.47197) | (2.29653) |
| Indep_ratio | −0.00647** | 0.09495* | 0.08637 |
| | (−2.03861) | (1.68670) | (1.53843) |
| Top3_pay | 0.00037 | 0.55907*** | 0.55957*** |
| | (1.33114) | (112.63806) | (113.04763) |
| Year | controlled | controlled | controlled |
| Industry | controlled | controlled | controlled |
| Constant | 0.82348*** | −8.34063*** | −7.24887*** |
| | (196.13607) | (−112.02469) | (−61.03991) |
| Observations | 24,708 | 24,708 | 24,708 |
| R-squared | 0.76387 | 0.48426 | 0.48715 |

Note:t-statistics in parentheses,

***$p < 0.01$,

**$p < 0.05$,

*$p < 0.1$

attention tend to possess stronger organizational resilience, which in turn helps to reduce the pay gap. As a crucial dynamic capability, organizational resilience enables companies to better adapt to external environmental changes and maintain operational stability. By emphasizing compliance, companies can mitigate the risk of non - compliance and enhance their ability to withstand potential crises. Consequently, this contributes to a more stable and equitable compensation structure, as employees are more likely to accept pay gaps when they perceive the company as resilient and proficient in risk management.

Incorporating compliance and social performance into a company's strategy is conducive to establishing a more equitable and sustainable remuneration structure. Therefore, companies should focus on strengthening their compliance culture and improving their social performance. This not only enhances their reputation and competitiveness but also ensures fair and transparent compensation practices.

## 7. Theoretical implications

This study contributes theoretically to understanding the interplay among management compliance attention, social performance, and pay gap in Chinese A-share listed firms. By integrating the Behavioral Theory and Equity Theory, we develop a comprehensive framework explaining how compliance-focused managerial attention and social performance collectively

shape compensation equity. The moderating role of social performance identified here extends Stakeholder Theory by demonstrating that CSR initiatives amplify the equity-enhancing effects of compliance systems.

Management compliance attention strengthens organizational resilience through improved risk governance and resource flexibility, which in turn mitigates pay inequality. This finding validates organizational resilience as a critical mediating mechanism linking compliance strategies to equitable outcomes, thereby expanding the application beyond traditional operational contexts.

Methodologically, this research enriches compliance theory by operationalizing management compliance attention through text analysis of MD&A sections. By examining compliance's impact on pay equity, we reveal compliance as a strategic resource capable of shaping organizational outcomes beyond mere regulatory adherence. This bridges cognitive perspectives on managerial attention with institutional theories of compliance, offering novel insights into how symbolic compliance practices translate into tangible equity improvements.

Overall, this study contributes to the emerging literature on responsible corporate governance by providing a nuanced understanding of the complex interplay between management compliance attention, social performance, and pay gap. By highlighting the mediating role of organizational resilience and the moderating effect of social performance, this study offers a more comprehensive theoretical framework for understanding how firms can achieve sustainable pay equity through effective compliance and social responsibility practices.

## 8. Practical implications

This study offers actionable insights for firms seeking to enhance pay equity and strengthen corporate governance and social responsibility practices. By understanding the relationships among management compliance attention, social performance, and pay gap, organizations can design targeted strategies to improve operational outcomes. Prioritizing compliance attention fosters operational transparency and fairness, enabling both management and employees to internalize legal and ethical standards. Integrating compliance into daily operations through structured guidelines and regular assessments mitigates non-compliance risks and strengthens organizational resilience.Transparent compensation systems aligned with performance metrics and organizational values are critical for maintaining employee satisfaction and operational stability. Firms that systematically review pay structures to reflect contributions and responsibilities can effectively manage pay gap perceptions, fostering cultures of equity. Engaging in community welfare, environmental sustainability, and employee well-being initiatives enhances corporate reputation and stakeholder trust. Regular stakeholder engagement provides feedback loops to align social initiatives with societal expectations.

Developing dynamic capabilities to adapt to market changes and external shocks is essential for organizational resilience. Investments in technology, innovation, and workforce development improve adaptability, while robust risk management frameworks (e.g., contingency planning) ensure stability during crises.Notably, ownership structure moderates the effectiveness of compliance and social performance initiatives. State-owned enterprises (SOEs) may derive greater equity benefits from these strategies, while private firms require targeted approaches to integrate compliance and social performance into core strategies. Leveraging these insights, firms can strategically enhance compliance cultures, optimize compensation systems, elevate social performance, and build resilience—actions that drive employee satisfaction, organizational performance, and sustainable business models.

## 9. Limitations and future research

This study's limitations suggest avenues for future inquiry. First, the exclusive focus on Chinese A-share listed firms from 2010–2022 may restrict generalizability. Replication in other institutional contexts or non-listed organizations could validate the findings' universality. Additionally, exploring industry-specific dynamics in the compliance attention-social performance-pay gap relationship would enhance contextual understanding.Second, operationalizing management compliance attention through annual report text analysis may not fully capture implementation effectiveness. Future studies could triangulate with qualitative data (e.g., management interviews, employee surveys) to deepen measurement validity.

Similarly, complementing ESG ratings with alternative social performance indicators (e.g., CSR expenditure, stakeholder engagement metrics) would provide a more comprehensive assessment.Third, while this study documents significant ownership-based variations in compliance-social performance effects, the underlying mechanisms remain underexplored. Future research could investigate how ownership structure influences compliance institutionalization and social performance integration processes, particularly in private firms where compliance may lack regulatory enforcement.Fourth, cross-cultural comparisons between China and other regions would illuminate how cultural values moderate the compliance-pay gap relationship. Such studies could address whether the observed effects persist in low-context cultures with different governance norms.

By addressing these limitations, future research can advance theoretical understanding of the compliance-social performance-pay gap nexus and inform context-specific governance strategies.

## 10. Conclusion and Policy implications

This study examines the impact of management compliance attention on pay gaps using a sample of Chinese A-share listed companies from 2010–2022. Findings reveal that both management compliance attention and social performance mitigate pay gaps, with social performance strengthening the negative impact of compliance attention. Organizational resilience partially mediates this relationship, enriching literature on pay gap determinants and corporate compliance outcomes.

Based on these findings, the following policy recommendations are proposed. First, firms should cultivate compliance cultures emphasizing integrity and fairness. Management compliance attention should be reinforced through structured training programs and awareness campaigns to ensure alignment with legal, regulatory, and internal policy frameworks in daily operations. Second, organizations should design transparent compensation systems aligned with equity principles, ensuring fairness in remuneration distribution processes. Third, social performance should be integrated into performance evaluation systems as a key metric for assessing managerial and departmental contributions. By linking executive compensation to social responsibility and sustainability outcomes, firms can incentivize engagement in societal initiatives, enhance overall social performance, and continuously refine governance mechanisms. Finally, policymakers and regulators should prioritize compliance capacity building in private enterprises, fostering social responsibility awareness and ensuring effective CSR implementation to align with stakeholder expectations.

## Supporting information

**S1 Data. Stata dataset (dta), do file (do) and empirical results (excel).**
(ZIP)

## Author contributions

**Conceptualization:** Fei Han, Yong Jiang.

**Data curation:** Yong Jiang.

**Methodology:** Yong Jiang.

**Project administration:** Fei Han, Yong Jiang.

**Writing – original draft:** Fei Han.

**Writing – review & editing:** Fei Han, Yong Jiang, Yanhan Sun.

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
