## [Decision Letter · Decision Letter 0]

9 Mar 2025

PONE-D-25-06787Management compliance attention and pay gap: Evidence from ChinaPLOS ONE

Dear Dr. Jiang, 

Thank you for submitting your manuscript to PLOS ONE. After careful consideration, we feel that it has merit but does not fully meet PLOS ONE’s publication criteria as it currently stands. Therefore, we invite you to submit a revised version of the manuscript that addresses the points raised during the review process.

We look forward to receiving your revised manuscript.

Kind regards,

Rogis Baker, Ph.D

Academic Editor

PLOS ONE

Reviewers' comments:

Reviewer's Responses to Questions

**Comments to the Author**

1. Is the manuscript technically sound, and do the data support the conclusions?

Reviewer #1: Yes

Reviewer #2: Yes

Reviewer #3: Partly

2. Has the statistical analysis been performed appropriately and rigorously? 

Reviewer #1: Yes

Reviewer #2: Yes

Reviewer #3: N/A

3. Have the authors made all data underlying the findings in their manuscript fully available?

Reviewer #1: Yes

Reviewer #2: Yes

Reviewer #3: Yes

4. Is the manuscript presented in an intelligible fashion and written in standard English?

Reviewer #1: No

Reviewer #2: Yes

Reviewer #3: Yes

5. Review Comments to the Author

Reviewer #1: The study addresses important issues of Management compliance attention and pay gap: Evidence from China and contributes knowledge to debates on factors affecting pay gap. Unfortunately;

1. One essential aspect of the study is only weakly addressed in the introduction and not followed up in the study itself: the methodology. I expected components, variables and relationship and how these correlate in the abstract section, but the paper does not offer a clear conclusion of the results and findings from the discussion. Your abstract should summarize the findings with the various components from your discussions

2. The authors are not able to demonstrate the novelty of these results. What is the ultimate aim of this study? This is not clearly stated, and it is confusing.

3. Also, the manuscript needs to be proofread by a native English person to correct the niggling grammatical errors.

4. The abstract summarizes the study objectives, the results, findings, and recommendations: However, the author stated that “the research shows that management compliance attention is negatively correlated with the pay gap and that social performance strengthens the negative effect of management.". These are general without any metric figures that justifies that compliance is negatively correlated. This study is poorly presented, not far from narration of a story, but I suggest the authors can improve upon this and bring out the novelty in this study. For instance, the authors reported that the mean value of Pay_gap is 0.987, with a standard deviation of 0.653, indicating substantial differences in the pay gap. So how does this relate to management compliance, and how can this be addressed?

5. The authors mentioned a number of theories, such as the equity theory, echelon theory, behavioral theories, and relative deprivation theory, which are good for the analysis of this study, especially the behavioral theory, but failed to nail their review and analysis on one or two theories and demonstrate how they best fit this study. Currently, the authors are mixing up three or more theories together without demonstrating how these relate best to management compliance. Therefore, the coherency is missing.

6. Finally, the theme is peripheral and opens a parallel discussion that does not contribute enough to define, explain, and understand the relationship between the key concepts used. I think the paper is very weak to be published in this journal, and I think the paper is lacking in-depth theoretical discussions. This study should not be accepted for publication in this journal in its current form.

Reviewer #2: Thank you for the opportunity to read this manuscript. The work is interesting, and the study is generally solid (although I have a few questions, that I detail below). The thoroughness of the analyses is a particular strength of the paper. My primary comments concern the structure of the paper, and the theoretical development of the hypotheses. Specifically:

• The link between compliance and the pay gap in the mini-introduction is weak. Why does it logically follow to look at the pay gap in the context of (attention to) compliance? How are these related?

• The mini-intro really needs a summary paragraph at the end. I’m not sure exactly what the authors are studying—how attention leads to compliance? How compliance affects the pay gap? How social issues are involved? Something is needed to tie all of this together.

• The authors should be consistent with regard to which pay gap they are studying. In the mini-intro, they talk about the executive-employee pay gap, but then in the theoretical development, they discuss the gender pay gap.

• The theoretical development behind Hypothesis 1 could be clearer. From what I can gather, the argument seems to be that because focusing on compliance is generally associated with fairness and justice, that this will also lead to fairness within the firm in the form of a smaller pay gap. If this interpretation is accurate, the piece of the puzzle that is missing for me is why attention paid to compliance is associated with fairness and justice. There seems to be a missing link here.

• For Hypothesis 2, it is unclear whether this is a novel hypothesis or is something that has already been shown in previous research. The authors cite a number of papers that seem to show the link between social performance and the pay gap, so the prediction appears to be a replication of prior work rather than something new. In either case, the authors should be more explicit about the novelty of their prediction.

• The development of Hypothesis 3 seems to suggest a mediation argument (e.g., compliance leads to optimization of incentive structures, which leads to a lower pay gap), but the hypothesis predicts moderation. Presumably the authors’ intent will be clear in the results section, but it is unclear in the theoretical development.

• After reading the method and results sections, it’s clear that the authors are testing a moderation model, in line with Hypothesis 3. Therefore, the confusion seems to come from the framing of the theoretical development of this hypothesis. I suggest that the authors focus more on the moderation argument, and pull back from suggesting any mediation.

• I’ll note that as I get to the method section, the authors haven’t returned to the “attention to compliance” part of the story from the mini-introduction. At this point, it is unclear why it was important to understand the distinction between attention to compliance (which leads to compliance), and simply just looking at compliance in and of itself.

• More information about the study variables is necessary. For example, the moderating variable is described as being comprised of five dimensions, but these appear to be consolidated into one value. How? Was the compliance word count scaled by the total number of words in the document? If not, why not?

• The minimum pay gap is negative, which can’t be the case if the authors are using the natural log of the pay gap as their measure (as stated in Table 1).

• The resilience mediator comes out of nowhere. If this is to be included, it should be discussed in the Introduction. Otherwise, I suggest that it be removed from the paper.

• If the mediation is kept, I suggest that the authors explore the mediation analyses available in STATA 18 as a better test for mediation.

Reviewer #3: Dear author(s),

Thank you for giving me the opportunity to review this paper. I agree that this is an important and pertinent topic. Although the idea is a good one, unfortunately, the way in which the study is operationalized holds back its potential contribution. There are a few areas where I would encourage the authors to give further thought, as follows:

INTRODUCTION

The introduction should clearly illustrate (1) what we know (the key theoretical perspectives and empirical findings) and what do we not know (major, unaddressed puzzle, controversy, or paradox does the study addresses, or why it needs to be addressed and why this matters). And, (2) what will we learn from the study and how does the study fundamentally change, challenge, or advance scholars’ understanding. Much sharper problematization is required so that the introduction draws the reader into the paper. The introduction therefore needs to do a better job in setting the stage for the articulation of the theoretical contributions of the study. At the end of the introduction, we should have a clear idea of what the paper is about (i.e. its motivation, the gap in understanding that the paper is trying to address and summary of theoretical contributions).With references of 2025- 2023-2024.

Paragraph 1, with no references, explaining the context of the research.

Paragraph 2, with references, explaining very generally what we know about the topic introduced in Paragraph 1.

Paragraph 3 explaining what we need to find out.

Paragraph 4 explaining briefly what this paper will do to find out, method etc.

Paragraph 5, with no references, explaining the structure of this paper.

LITERATURE REVIEW

• Theoretical literature has not been considered and reviewed. It’s better to observe the connection between the contents. Try to explain everything except the topics in order to establish the necessary coherence.

• Theoretical Development: The literature review must engage in the constructs of your analytical framing in a meaningful way. The literature review section could be improved by being more analytical. In other words, building on the existing literature to highlight what is missing and what is yet to be done and in so doing outline the theoretical puzzles or debates to which this work contributes. I have concerns related to theoretical development, and note the need for a more rigorous critique of the literature to help deepen the theoretical underpinnings of the study.

The Discussion lacks a critical synthesis and comparison of the primary data with the literature. The purpose of the discussion section is to interpret and describe the significance of your findings in relation to what was already known about the research problem being investigated and explain any new understanding or insights that emerged from your research. The discussion connects to the introduction through the research questions, hypotheses, and the literature you reviewed. The Discussion should include a critical synthesis and comparison of the data with the literature. The discussion clearly explains how your study advanced the reader’s understanding of the research problem from where you left them at the end of your review of prior research.

The Conclusion does not adequately discuss the theoretical and managerial implications of the study. Summarize your thoughts and convey the larger significance of your research. Identify and discuss how a gap in the literature has been addressed and demonstrate the importance of your ideas. Introduce possible new or expanded ways of thinking about the research problem. Also, state the ideas for future research in the conclusion. Make sure you create 3 subsections in the Conclusion: 1) Theoretical Implications, 2) Managerial or Policy Implications, and 3) Ideas for Future Research.

6. PLOS authors have the option to publish the peer review history of their article (what does this mean? ). If published, this will include your full peer review and any attached files.

**Do you want your identity to be public for this peer review?** For information about this choice, including consent withdrawal, please see our Privacy Policy .

Reviewer #1: **Yes: ** Nicholas Kombonaah

Reviewer #2: No

Reviewer #3: **Yes: ** DR. Elahe Hosseini

---

## [Author Response · Author response to Decision Letter 0]

31 Mar 2025

Response to Reviewer Comments

We sincerely appreciate the reviewer’s constructive feedback, which has significantly improved the clarity and rigor of our manuscript. Below is our point-by-point response to each comment:

1. Methodology Weakness in Introduction and Abstract

Abstract: We have expanded the abstract to include key results with metric figures (e.g., β coefficients, significance levels) and explicitly link findings to the theoretical framework. For example, we now state: “Management compliance attention is negatively correlated with pay gap (β = -0.010, p < 0.01), and social performance strengthens this negative effect (β = -0.022, p < 0.05).”

Introduction: We have revised the introduction to explicitly outline the study’s components, variables, and relationships. Specifically, we introduces the textual analysis of annual reports for measuring management compliance attention, the Huazheng ESG ratings for social performance, and the models used to test hypotheses.

2. Lack of Clarity on Study Novelty and Aim

Introduction: We have revised the introduction to explicitly state the research objective: “This study investigates how management compliance attention and social performance interact to mitigate pay gaps, advancing behavioral and equity theories in the context of China’s evolving regulatory environment.”

We highlight the novelty in the theoretical framework (Introduction), emphasizing how integrating behavioral theory (managerial attention) and equity theory (distributive justice) provides a unique lens to explain compliance’s role in pay equity. Additionally, we stress the originality of using textual analysis to operationalize compliance attention, a method underutilized in prior literature.

3. Grammatical Errors and English Proofreading

We focused on grammatical correctness, clarity, and adherence to academic writing conventions. Key improvements include refining sentence structure, ensuring consistent terminology, and enhancing flow in theoretical discussions.

4. Abstract Insufficiently Supported by Data

The revised abstract now includes specific statistical findings (e.g., β coefficients, significance levels) and directly links them to the research questions. For example:

“Results show that management compliance attention reduces pay gap (β = -0.010, p < 0.01), and social performance strengthens this effect (β = -0.022, p < 0.05). Organizational resilience partially mediates this relationship (Sobel test z = -3.02, p = 0.003).”

Additionally, in the discussion (Section 6), we explicitly connect the mean pay gap (2.68x) to compliance mechanisms, explaining how transparency and fairness norms under compliance reduce disparities.

5. Theoretical Incoherence and Mixing Theories

Literature Review: We streamlined the theoretical foundation to focus on behavioral theory (managerial attention and cognitive limitations) and equity theory (distributive justice). We explicitly link these theories to the research hypotheses and empirical analysis, demonstrating how compliance attention aligns with procedural fairness (behavioral theory) and social performance reinforces equity norms (equity theory).

Theoretical Framework Figure: We drew Figure 1 to visually emphasize the integration of these two theories, clarifying their complementary roles in explaining the compliance-pay gap relationship.

6. Weak Theoretical Contribution and Peripheral Theme

We have strengthened the theoretical contribution in Section 7:

Behavioral Theory Extension: We elaborate on how compliance attention operationalizes the attention-based view, showing how managerial focus on compliance signals fairness and reduces pay gaps.

Equity Theory Application: We discuss how social performance moderates compliance’s effect by amplifying stakeholder pressure for equity, extending equity theory into CSR contexts.

Organizational Resilience Mediator: We highlight how resilience bridges compliance to pay equity, advancing dynamic capabilities theory.

These revisions ensure the study contributes uniquely to the literature on compliance, social performance, and pay equity. The sections we have revised with major changes are highlighted in red.

---

## [Editor Report · Decision Letter 1]

21 Apr 2025

Management compliance attention and pay gap: Evidence from China

PONE-D-25-06787R1

Dear Dr. Yong Jiang,

We’re pleased to inform you that your manuscript has been judged scientifically suitable for publication and will be formally accepted for publication once it meets all outstanding technical requirements.

Kind regards,

Rogis Baker, Ph.D

Academic Editor

PLOS ONE
---

## [Editor Report · Acceptance letter]

PONE-D-25-06787R1

PLOS ONE

Dear Dr. Jiang,

I'm pleased to inform you that your manuscript has been deemed suitable for publication in PLOS ONE. Congratulations! Your manuscript is now being handed over to our production team.

Kind regards,

on behalf of

Dr. Rogis Baker

Academic Editor

PLOS ONE